# Pharmacogenetic Testing in Acute and Chronic Pain: A Preliminary Study

**DOI:** 10.3390/medicina55050147

**Published:** 2019-05-16

**Authors:** Lorenzo Panella, Laura Volontè, Nicola Poloni, Antonello Caserta, Marta Ielmini, Ivano Caselli, Giulia Lucca, Camilla Callegari

**Affiliations:** 1Department of Rehabilitation, ASST Gaetano Pini—CTO, Via Isocrate 19, 20122 Milan, Italy; lorenzo.panella@asst-pini-cto.it (L.P.); laura.volonte@asst-pini-cto.it (L.V.); antonellovalerio.caserta@asst-pini-cto.it (A.C.); 2Department of Medicine and Surgery, Division of Psychiatry, University of Insubria, Viale Luigi Borri 57, 21100 Varese, Italy; nicola.poloni@uninsubria.it (N.P.); marta.ielmini@hotmail.it (M.I.); ivo.ivo@aliceposta.it (I.C.); giulia1lucca@gmail.com (G.L.)

**Keywords:** pain, analgesic, pharmacogenetic testing, pharmacological therapy, effectiveness, adverse effects

## Abstract

*Background and Objectives:* Pain is one of the most common symptoms that weighs on life’s quality and health expenditure. In a reality where increasingly personalized therapies are needed, the early use of genetic tests that highlights the individual response to analgesic drugs could be a valuable help in clinical practice. The aim of this preliminary study is to observe if the therapy set to 5 patients suffering of chronic or acute pain is concordant to the Pharmacogenetic test (PGT) results. *Materials and Methods:* This preliminary study compares the genetic results of pharmacological effectiveness and tolerability analyzed by the genetic test Neurofarmagen Analgesia^®^, with the results obtained in clinical practice of 5 patients suffering from acute and chronic pain. *Results:* Regarding the genetic results of the 5 samples analyzed, 2 reports were found to be completely comparable with the evidences of the clinical practice, while in 3 reports the profile of tolerability and effectiveness were partially discordant. *Conclusion:* In light of the data not completely overlapping with results observed in clinical practice, further studies would be appropriate in order to acquire more information on the use of Neurofarmagen in routine clinical settings.

## 1. Introduction

Pain is the most common factor motivating healthcare use, as well as one of the main health care system spending factors. In particular, chronic pain is such a diffuse and disabling condition that it is considered a syndrome and not merely a symptom [1,2]. Individual sensitivity and pain perception, as well as antalgic treatment response, are influenced by numerous factors such as duration, cultural difference, weight, age, co-morbidity, concomitant therapies, psychological factors, and genetic predisposition [3,4,5]. Pain, especially chronic pain, includes a wide range of treatments ranging from anti-inflammatory and simple analgesics to major opioids, cannabinoids, antidepressants, antipsychotics, local anesthetics, and ketamine [6,7,8,9]. Moreover, the method of pain assessment influences acute pain diagnosis which may lead to further chronicity of underdiagnosed pain. One of the potential reasons for variability in observed clinical response to pain may be related to the method of assessment and intensity measurement for different patient populations. For example, inadequate pain evaluation may be seen in postoperative patients (assessment of pain at rest, as opposed to pain after movement) [10] and in critically ill or non-verbal patients (inadequate use of behavioral pain scales) [11].

In recent years an increased interest in the genetic and epigenetic correlates of pain was observed both for the genetic origin of pain’s sensitivity threshold, and for the individual response to analgesic treatments [12,13]. A better understanding of individual sensitivity to drugs would allow a more patient-targeted approach with consequent reduction of the antalgic response time, reduction of failed treatment attempts, a remarkable reduction of healthcare costs, and therefore a more rapid improvement in quality of life [14]. Some individuals may be less responsive to therapeutic pharmacological analgesic treatments, while others may be unresponsive or exhibit adverse events. Consequently, the knowledge of individual responsiveness to antalgic drugs, besides allowing one to reach the predetermined therapeutic objective more quickly, may also reduce adverse effects [13]. The main genetic modifications implicated in different pain responses to analgesics seem to be related to drug metabolism. Genes coding for enzymes, many of which belong to the family of cytochrome P-450 with its multiple isoforms (CYP2D6, CYP2C9, CYP2C8, COMT, OPRM1) often present individual variability due to single nucleotide polymorphisms (SNPs), as confirmed by the literature [15]. The CYP2D6 gene, located on chromosome 22 and coding for a member of the cytochrome P450 enzyme superfamily (cytochrome P450, family 2, subfamily D, polypeptide 6), is responsible for the metabolism of about 25% of the drugs currently used in clinical practice. CYP2D6 is highly relevant to the metabolism of minor and major opioids [16] and for several antidepressant drugs. Several variants cause a loss of CYP2D6 function, and the homozygous subjects for these variants are called “slow metabolizers” (SM); they present, for the same dose of drug, higher plasma levels than normal [17]. The CYP2C9 gene is located on chromosome 10 that codes for a member of the cytochrome P450 enzyme superfamily (cytochrome P450, family 2, subfamily C, polypeptide 9). CYP2C9 is responsible for the metabolism of 15% of commercially available drugs. Several non-steroidal anti-inflammatory drugs (NSAIDs) are metabolized by CYP2C9 and, in some cases, the deficiency of this enzyme has been associated with an increased risk of gastrointestinal hemorrhage in patients treated with NSAIDs [18]. The CYP2C8 gene is located on chromosome 10, which codes for a member of the cytochrome P450 enzyme superfamily (cytochrome P450, family 2, subfamily C, polypeptide 8). Together with other enzymes, such as CYP2C9, CYP2C8 intervenes in the metabolism of NSAIDs, including ibuprofen and diclofenac. The gene has a variant that has been associated with a reduction in drug clearance and an increased risk of gastrointestinal hemorrhage. The COMT gene is located on chromosome 22 that encodes the catechol-O-methyltransferase enzyme, whose function is the degradation of the dopamine and norepinephrine catecholamines and the consequent modulation of the levels of these neurotransmitters in inter-synaptic space. Various polymorphisms with influence on COMT activity are known; for example, the Val158Met polymorphism has been associated with the μ opioid system response and the efficacy of morphine treatment both on acute and chronic pain [19]. The OPRM1 gene is located on chromosome 6 that encodes for μ1, the main opioid receptor target of endogenous opioid peptides (such as β-endorphin and enkephalins) and opioid analgesics (such as morphine). Several studies indicate that post-operative and oncological patients who carry at least one copy of the G allele of the gene have a lower pain threshold and experience an increase in opioid use [20]. Moreover, this polymorphism has also been associated with the susceptibility to opioid and alcohol dependence [14]. In most cases the therapeutic choices of clinicians regarding analgesic drugs are empirical, and the different individual response may represent a further step to overcome in order to obtain an adequate analgesic response. Several attempts are in fact often necessary before obtaining adequate analgesia. This leads to longer response times, greater patient distress and higher healthcare costs [21]. The rationale for this study is based on the possible clinical utility of early identification of genetic polymorphisms responsible for the different individual responses of analgesic drugs.

The aim of this preliminary study is to observe if the therapy set to 5 patients suffering of chronic or acute pain is concordant to the pharmacogenetic test (PGT) results.

## 2. Materials and Methods

### 2.1. Sample

This naturalistic study had foreseen a preliminary assessment of 5 patients affected by acute and chronic pain referring to Gaetano Pini Orthopedic Institute. Patients had to fulfill the following inclusion criteria: to be older than 18 years; to suffer from acute or chronic pain; to subscribe a generic written informed consent.

Patients were evaluated during an outpatient visit at the Gaetano Pini Institute. All patients were already taking pain medication at the time of assessment. The genetic test was performed by two investigators who did not know the clinical response at the therapy, while clinicians were not aware of the test results, carried out subsequently.

All patients provided a general written informed consent for processing personal data as part of the routine diagnostic assessment procedure and quality check processes. Assessment tools and scales were administered by clinicians at the patient’s facilities, as part of clinical routine practice. As patient data was made anonymous, obscuring sensitive information used in the research to protect the recognizability of the patients, according to the Italian legislation (D.L. 196/2003, art. 110 - 24 July 2008, art. 13), the Provincial Health Ethical Review Board, consulted prior to the beginning of the study, confirmed that it did not need authorization from the Board. The study was carried out in accordance with the ethical principles of Declaration of Helsinki (with amendments) and Good Clinical Practice.

### 2.2. Genetic Analysis

The genetic analysis was performed by the Neurofarmagen Analgesia^®^ kit. This test scans the genetic polymorphisms linked to the pharmacokinetics of analgesic drugs belonging to the NSAIDs, opioids, antidepressants, antiepileptic, triptans, 5HT-3 receptor antagonists, and ergotamine’s derivatives. The test examines the isoforms of cytochrome CYP2B6, CYP2C8, CYP2C9, CYP2C19, CYP2D6, and CYP3A4, which are responsible for the metabolism of each of the drugs considered. The test allows us to determine the patient response both to peripheral analgesics like NSAIDs and cox-2, and to central analgesics as paracetamol, tricyclics, serotonin reuptake inhibitors (SSRI), anticonvulsants, opioids, and triptans. The genetic test’s administration is carried out by taking a patient’s saliva sample in an amount equal to about 1 ml. Given the same possibility to obtain the genetic information from biological material, use of saliva samples allow for greater practicality in the execution of the test, and it is less invasive. DNA was extracted from patient saliva samples with the genomic DNA Isolation kit (Norgen Biotek Corp. Thorold, ON, Canada). DNA quality was evaluated by microvolume spectrometry. The sample is collected by the patient under the doctor’s guidance through the kit provided and remains stable at room temperature for a maximum of 15 days, the time required for sending it off to the laboratory to be analyzed. During the 30 min before sample collection the patient should not consume food, drink or chewing gum, must not smoke and must have completely removed the presence of any lipstick. The results are available within 10 working days from the date of arrival of the sample at Neurofarmagen laboratory (AB-BIOTICS, Barcelona, Spain) which is declared by the company as having the necessary authorization to operate as a health laboratory using biological samples. Genotyping of single nucleotide polymorphisms was performed by OpenArray^®^ Technology on the QuantStudio™ 12K Flex Real-Time PCR System (Thermo Fisher Scientific Inc., Waltham, MA, USA) using a custom designed array. Saliva samples were marked with an electronic numerical barcode associated with the patient, and the identification code was archived at the Gaetano Pini Orthopedic Institute in Milan. The archive in question contains the correspondences necessary to associate the clinical and demographical data of each patient with the Neurofarmagen genetic record. In that way no sample was sent in association of personal data. The collected data have been used exclusively for research purposes. The test evaluates 6 different genes, putting them in relation to active 48 drugs. For each patient the report provides a table in which the molecule evaluated correspond to a color coding: (1) green—expectation of higher likelihood of the good response to treatment or a good tolerability profile than average; (2) white—index of a standard response, not different from the general population; (3) yellow—requiring more careful dose monitoring; and (4) red—for high risk of adverse effects or not expected efficacy.

### 2.3. Clinical Assessment

Patients were evaluated with algometric scales:McGill Pain Questionnaire-Short Form—MGPQ-SF [22]: this is a rating scale developed at McGill University by Melzack and Torgerson in 1971 and it is a self-report questionnaire that allows patients to give a good description of the quality and intensity of the experienced pain. It is composed of 78 words in 20 sections that are related to pain. Patients have to mark the words that best describe their pain (multiple markings are allowed). Different sections correspond to different pain’s features: Sensory (sections 1–10), Affective (sections 11–15), Evaluative (section 16), and Miscellaneous (sections 17–20).Visual Analogue Scale—VAS [23]: the VAS scale consists of a strip of 10 cm paper which at the ends has two end points which are defined with 0 (no pain) and 10 (the worst pain that I can imagine). Patients have to mark the pain at a point on the scale as subjectively perceived at the moment of evaluation.Numerical Rating Scale—NRS [23]: the NRS is a 11-point scale for patient self-reporting of pain. The score is between 0 (no pain) and 10 (severe pain) and expresses the intensity of the pain perceived subjectively at the moment of evaluation.

## 3. Results

Clinical characteristics of the patients and treatments prescribed are shown in Table 1. The first patient (P1) was a 45-year-old woman suffering from fibromyalgia not responsive to diclofenac, paracetamol and then major opioids. The result of the test suggested for diclofenac and paracetamol a standard response without a need for dosage adjustment; the PGT highlighted a warning for the use of minor opioids (yellow response, due to an intermediate metabolism), while suggesting the possibility of a standard response to some major opioids.

The second patient (P2) was a 22-year-old woman suffering from chronic pain resulting from Ewing’s femoral sarcoma resection and mega prosthesis implant replacement at the age of 16. The patient had taken several different analgesic molecules (good response to nimesulide, ineffectiveness to paracetamol, gastrointestinal intolerance to naproxen and ibuprofen, effectiveness to buprenorphine e tramadol, effectiveness to pregabalin). In this case a substantial correspondence between the report provided by the genetic analysis and effectiveness and tolerability profile observed in clinical practice was found.

The third patient (P3) was a 45-year-old woman with a diagnosis of end stage renal failure in systemic lupus erythematosus (SLE) with contraindication to the use of NSAIDs and a short-term program of renal transplantation. The patient underwent a laparoscopic abdominal operation with analgesia performed with a high dose of tramadol and without side effects, while in the post-transplant a low dose of morphine and subsequently 2 mg/day of Paracetamol for 5 days, were successfully used. Regarding tramadol, the test indicated the patient as a slow metabolizer (CYP2D6) and recommended reducing the dose by 30%, along with paying particular attention to the occurrence of adverse effects such as nausea, vomiting, constipation, respiratory depression, confusion, and urinary retention, or to choose an alternative drug (yellow response). The drug was instead used at maximal doses intravenously with complete response to pain and without collateral or treatment emergent adverse events—TEAE (emesis, as potential side effect, was not evaluated for overlapping adequate coverage with metoclopramide). With regard to the use of morphine, the test showed the presence of a variant of COMT and OPRM1 that resulted in a reduced response to opioid agonists: the patient may have needed higher dosage or more frequent administration of the drug for adequate pain control. The patient was instead subjected to post-operative therapy with morphine at minimum therapeutic doses with optimal and rapid response. With regard to paracetamol, the PGT indicated a standard response, meaning a risk of undesirable effects not higher than general population. The patient, instead, showed signs of severe hepatic toxicity as a result of maximum therapeutic antalgic dose for 5 days of treatment. The situation required clinical and instrumental hepatological monitoring; the normalization of the liver enzyme required about 60 days. In this case some discordances between the clinical setting and the PGT’ suggestion emerged.

The fourth patient, (P4) was a 75-year-old woman affected by persistent osteoarthritis (OA) pain poorly responsive to common NSAIDs. The patient was clinically known as non-responsive to NSAIDs and discreetly responsive to paracetamol-codeine combination without evidence of significant adverse effects at a common therapeutic dosage. The result of the genetic test was indicative for standard response to NSAIDs and highlighted the need for specific dose monitoring and/or lower probability of positive response to codeine (yellow response). The PGT indicated the patient as an intermediate metabolizer of codeine due to the CYP2D6 gene. For the same patient, the test suggested the possibility of using major opioids (with specific reference to morphine).

The last patient (P5) was a 55-year-old man suffering from chronic pain caused by early OA. The patient presented an excellent response to the analgesic drugs used (paracetamol-codeine and NSAIDs) at the doses commonly used in clinical practice. In this case the genetic test showed a standard profile of efficacy and tolerability for the molecules used, confirming the clinical level.

Total scores of algometric scales for pain evaluation are reported for each patient in Table 2.

## 4. Discussion

Pain is a syndrome that has a major impact on life’s quality. It is intended as a complex phenomenon characterized by a huge subjective component and dependent on different endogenous and external factors [24,25]. An early investigation of subjective sensitivity to drug therapies could be a useful instrument in clinical practice. Over time, genetic testing has become more accessible and less expensive. However, currently available data are scarce and heterogeneous, and there are few clear evidences that genetic tests are useful tools in clinical practice [26,27,28,29,30].

Pain treatment is a significant challenge for clinicians and could hesitate in poor outcome, particularly among patients treated with multiple drugs, in complications and adverse drug effects [31,32], no improvement of pain, or even worsening of it, and decline in functional status. So, more accurate information could be useful in routine clinical practice. Moreover, pain therapies should be monitored regularly not only for efficacy, but also for adverse effects and addiction. In this panorama, genetic testing could be a valid tool in current practice if used in an appropriate way [30]. In pain measurement, different variables should be taken into consideration; cognitive profile of the patient, physical limitations of the patient, psychiatric comorbidity or even simulation, ethnic and cultural variations could play an important role [33,34,35,36,37,38]. In light of these elements, the use of pharmacogenetic testing could be a useful contribution to objective measurement of response in pain pharmacotherapy [30]. Our preliminary results seem to be miscellaneous. In fact, the genetic test was concordant with the clinicians’ choice in two cases, but not completely overlapping in the other three cases. While in patient 2 and patient 5 the efficacy and tolerability profile provided by the test corresponded to the clinical response, in patient 1 and patient 4 the test seems to be discordant to the clinical practice. In patient 3, the PGT would have oriented the therapeutic choice quite differently from what done in clinical setting. Lastly, it is important to point out that in no case the results of genetic tests were red or green, but only standard or yellow, excluding both the higher effectiveness of the drugs and the high risk of adverse effects. Therefore, the test could be interpreted as not very sensitive but, in any case, it is evident that the sample presented is too heterogeneous and numerically limited in order to consider our descriptive analysis as indicative. Furthermore, possible pharmacodynamic and pharmacokinetic interferences related to comorbidity with other pathologies are not considered in the present study, but available in the extensive report of the pharmacogenetic test. For example, the elderly patient suffering from arthrosis assumed, at the time of recruitment, antihypertensive and antidiabetic drugs while the young patient who was recovering from a renal transplantation, took immunosuppressive anti-rejection therapies and this could be a confounding factor. The young female patient, suffering from a chronic pain syndrome consequent to the implant of femoral prosthesis, has a positive history of emotional disorder and, as confirmed by the literature, it is known that chronic pain and psychological discomfort can influence each other, also leading to a worsening of pain and of the pharmacological response [39].

Limitations of this study include the small sample size due to the pilot nature of the study and a lack of control group. The measures presented are primarily descriptive.

There is no doubt that genetic variations that involve gene coding for enzymes involved in drug metabolism, absorption, distribution, and elimination can significantly modify the therapeutic effects and tolerability profile of the drugs, and that knowing these polymorphisms would potentially allow us to customize therapy, maximize therapeutic efficacy, and minimize unwanted or adverse effects [40,41,42,43,44,45,46]. From the experience reached in the use of the Neurofarmagen test, it would seem that it shows a greater efficacy in predicting the side effects and need for dosage adjustment of analgesic drugs rather than therapeutic efficacy on pain symptoms. Pain is in fact greatly subjective and depends on several factors, not just genomics, so that a genetic analysis may be insufficient to evaluate this symptom in its complexity. In light of this, it seems to be desirable to intensify the studies on this subject in order to have a larger record that can be usefully utilized to help clinicians make more targeted and ultimately more effective therapeutic choices.

## Figures and Tables

**Table 1 medicina-55-00147-t001:** Clinical characteristics of the patients and treatments prescribed.

Patient	Sex	Age	Diagnosis	Comorbidity	Therapy	Test Results
P1	F	45	Widespread pain (fibromyalgia)	Gastritis; irritable colon syndrome; depression	Aceclofenac	Not evaluated
					Diclofenac	Standard response
					Paracetamol	Standard response
P2	F	20	Chronic pain resulting from Ewing’s femoral sarcoma resection and mega prosthesis implant replacement at the age of 16	Iatrogenic pulmonary restrictive syndrome caused by chemotherapy; dysthymia	Nimesulide	Not evaluated
					Paracetamol	Standard response
					Naproxen	Not evaluated
					Ibuprofen	Yellow response (increased risk of side effects)
					Buprenorphine	Standard response
					Tramadol	Yellow response (lower efficacy)
					Pregabalin	Standard response
P3	F	45	End stage renal failure with contraindication to the use of NSAIDs and a short-term program of renal transplantation	Systemic lupus erythematosus in remission	Paracetamol	Standard response
					Tramadol	Yellow response (increased risk of side effects)
P4	F	75	Persistent OA pain	Arterial hypertension; hypoacusis; recurrent and mild gastritis; irritable colon syndrome; subclinical thyroiditis	Several kinds of NSAIDs	Standard response
					Tramadol	Yellow response (lower efficacy)
					Buprenorphine	Standard response
					Codeine	Yellow response (lower efficacy)
					Paracetamol	Standard response
P5	M	55	Chronic pain caused by early OA	None	Several kinds of NSAIDs	Standard response
					Paracetamol	Standard response
					Codeine	Standard response

F: female, M: male, NSAID: non-steroidal anti-inflammatory drugs, OA: osteoarthritis.

**Table 2 medicina-55-00147-t002:** Total score of algometric scales McGill Pain Questionnaire-Short Form (MGPQ-SF), Numerical Rating Scale (NRS) and Visual Analogue Scale (VAS) representative of each patient.

Patient	MGPQ-SF	NRS	VAS
Patient 1 (P1)	45	8	8.07
Patient 2 (P2)	24	6	5.4
Patient 3 (P3)	7	0	0.43
Patient 4 (P4)	30	9	4.08
Patient 5 (P5)	26	5	4.7
Mean	26.4	5.6	4.69
Standard Deviation (SD)	13.61	3.5	2.74

MGPQ-SF: McGill Pain Questionnaire-Short Form, NRS: Numerical Rating Scale, VAS: Visual Analogue Scale.

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
