# Peer review of "Pharmacogenetic Testing in Acute and Chronic Pain: A Preliminary Study"

_medicina, 2019, doi:10.3390/medicina55050147_

Round 1

Reviewer 1 Report

Thank you for giving me the opportunity to review a manuscript titled: “Pharmacogenetic Testing in Acute and Chronic Pain: A Preliminary Study” written by Panella et al. The task undertaken by the authors is a very important one and the authors’ effort should be appreciated. The manuscript is well-written and concise, yet some points need highlighting and some improvement.

I would suggest adding an important element early in the Introduction section, stating that the method of pain assessment influences acute pain diagnosis and this may lead to further chronicity of underdiagnosed pain. One of the potential reasons for variability in observed clinical response to pain may be related to the method of assessment and intensity measurement for different patient populations. Inadequate pain evaluation may be seen in postoperative patients (assessment of pain at rest as opposed to pain after movement) 1 and in critically ill or non-verbal patients (inadequate use of behavioral pain scales) 2:

1. Gilron I, Vandenkerkhof E, Katz J, Kehlet H, Carley M Evaluating the Association Between Acute and Chronic Pain After Surgery: Impact of Pain Measurement Methods. Clin J Pain. 2017 Jul; 33(7):588-594.

2. Kotfis K, Zegan-Barańska M, Szydłowski Ł, Żukowski M, Ely EW. Methods of pain assessment in adult intensive care unit patients - Polish version of the CPOT (Critical Care Pain Observation Tool) and BPS (Behavioral Pain Scale). Anaesthesiol Intensive Ther. 2017;49(1):66-72. doi: 10.5603/AIT.2017.0010.

The refences regarding individual phenotypes should appear after each of them. Please be more specific regarding analgesic medications metabolism and consider citing:

1. St Sauver JL, Olson JE, Roger VL, et al. CYP2D6 phenotypes are associated with adverse outcomes related to opioid medications. Pharmgenomics Pers Med. 2017;10:217-227. Published 2017 Jul 24. doi:10.2147/PGPM.S136341

2. Figueiras A, Estany-Gestal A, Aguirre C, et al. CYP2C9 variants as a risk modifier of NSAID-related gastrointestinal bleeding: a case-control study. Pharmacogenet Genomics. 2016;26(2):66-73.

Moreover, I think that each isoform of cytochrome P-450 examined by the Neurofarmagen Analgesia kit should be described briefly in the introduction (their substrates).

In the Methods section the Authors wrote that “The NRS is a 10-point scale…”, please correct as this is an 11-point scale.

In the Results section - please be consequent with time – “The first patient (P1) is….”, yet the authors write - “The fourth patient (P4) was…”.

In the Results section– Patients no. 1 has the poorest description of the past medical history and no information regarding co-morbidities, yet her MGPQ-SF and VAS results were the highest. The pre-test history should be more detailed, as there seems to be something missing.

In the Results section the results are described, yet the details of the test have not been shown. There should be a detailed table regarding genetic testing with Neurofarmagen Analgesia kit for each patient to show contrast to clinical evaluation.

Author Response

Reviewer 1

Thank you for giving me the opportunity to review a manuscript titled: “Pharmacogenetic Testing in Acute and Chronic Pain: A Preliminary Study” written by Panella et al. The task undertaken by the authors is a very important one and the authors’ effort should be appreciated. The manuscript is well-written and concise, yet some points need highlighting and some improvement.

We are obliged for your time to review our paper and for your precious suggestions. Response to your comments are provided below in red.

I would suggest adding an important element early in the Introduction section, stating that the method of pain assessment influences acute pain diagnosis and this may lead to further chronicity of underdiagnosed pain. One of the potential reasons for variability in observed clinical response to pain may be related to the method of assessment and intensity measurement for different patient populations. Inadequate pain evaluation may be seen in postoperative patients (assessment of pain at rest as opposed to pain after movement) 1 and in critically ill or non-verbal patients (inadequate use of behavioral pain scales) 2:

1.      Gilron I, Vandenkerkhof E, Katz J, Kehlet H, Carley M Evaluating the Association Between Acute and Chronic Pain After Surgery: Impact of Pain Measurement Methods. Clin J Pain. 2017 Jul; 33(7):588-594.

2.      Kotfis K, Zegan-Barańska M, Szydłowski Ł, Żukowski M, Ely EW. Methods of pain assessment in adult intensive care unit patients - Polish version of the CPOT (Critical Care Pain Observation Tool) and BPS (Behavioral Pain Scale). Anaesthesiol Intensive Ther. 2017;49(1):66-72. doi: 10.5603/AIT.2017.0010.

We added a paragraph in the Introduction section on the influence of the method of pain assessment on acute pain diagnosis and related potential reasons for variability in observed clinical response to pain, citing the references suggested.

The refences regarding individual phenotypes should appear after each of them. Please be more specific regarding analgesic medications metabolism and consider citing:

-          1. St Sauver JL, Olson JE, Roger VL, et al. CYP2D6 phenotypes are associated with adverse outcomes related to opioid medications. Pharmgenomics Pers Med. 2017; 10:217-227. Published 2017 Jul 24. doi:10.2147/PGPM.S136341

-          2. Figueiras A, Estany-Gestal A, Aguirre C, et al. CYP2C9 variants as a risk modifier of NSAID-related gastrointestinal bleeding: a case-control study. Pharmacogenet Genomics. 2016;26(2):66-73.

We reported the references regarding individual phenotypes after each of them in the Introduction section. We specified better the analgesic medications metabolism citing the suggested references.

Moreover, I think that each isoform of cytochrome P-450 examined by the Neurofarmagen Analgesia kit should be described briefly in the introduction (their substrates).

We described briefly each isoform of cytochrome P450 examined by the Neurofarmagen Analgesia kit in the introduction.

In the Methods section the Authors wrote that “The NRS is a 10-point scale…”, please correct as this is an 11-point scale.

We corrected the sentence (11-point scale instead of 10-point scale) as suggested.

In the Results section - please be consequent with time – “The first patient (P1) is….”, yet the authors write - “The fourth patient (P4) was…”.

We corrected the verbal tenses, as suggested.

In the Results section– Patients no. 1 has the poorest description of the past medical history and no information regarding co-morbidities, yet her MGPQ-SF and VAS results were the highest. The pre-test history should be more detailed, as there seems to be something missing.

We clarified the patient 1 description adding the details on the past medical history and information about comorbidities.

In the Results section the results are described, yet the details of the test have not been shown. There should be a detailed table regarding genetic testing with Neurofarmagen Analgesia kit for each patient to show contrast to clinical evaluation.

We provided the genetic testing results for each patient as supplement material. We include a new table (Table 1) showing clinical characteristics of patients, diagnosis, comorbidities, treatments prescribed and pharmacogenetic test results.  

Reviewer 2 Report

This study by Panella and colleagues reported the data from a preliminary study. The authors attempted to find the association between the clinical response to analgesic drugs and pharmacogenetic testing in patients suffering the chronic or acute pain. They performed the genetic analysis to evaluate the isoforms of different targets in the saliva samples from 5 patients using a commercial kit. Then they assessed the clinical responses to different pain drugs and tried to find a connection between the results from two different experiments. It is very interesting and important to find some genetic evidence to explain the various response to the analgesic drugs, and the findings will provide some guidance for personalized medication. Unfortunately, the results from this study found that only 2 out of 5 patients showed some level of correlation between the results genetic analysis and clinical assessment.

        Overall, this study is interesting with a lot of limitations. Major revisions need to be done before of whether or not be accepted for publishing.

Major Essential Revisions

1)      This is human study and have to have approved protocol from Institutional Research Board for publishing the data in United States. However, it might be different in different country.

2)      What is the rational for using the saliva samples for genetic analysis?

3)      Why do the samples need to stay at room temperature for 15 days? Most of the protein would degrade during the time.

4)      The authors have to provide the details about how to perform the genetic analysis.

5)      The authors have to provide the results of genetic analysis of patient #1, 2, and 5 in detail.

6)      Missing citations through the paper. No references from Line 49-77,  Line 137-148.

Minor Revisions

1)      In Line 19, “5 tests, 2 tests” should be “5 samples, 2 samples”

2)      Need the product information for the Neurofarmagen Analgesia Kit

3)      In the Table1, the “Media” should be “Mean”

Author Response

Reviewer 2

This study by Panella and colleagues reported the data from a preliminary study. The authors attempted to find the association between the clinical response to analgesic drugs and pharmacogenetic testing in patients suffering the chronic or acute pain. They performed the genetic analysis to evaluate the isoforms of different targets in the saliva samples from 5 patients using a commercial kit. Then they assessed the clinical responses to different pain drugs and tried to find a connection between the results from two different experiments. It is very interesting and important to find some genetic evidence to explain the various response to the analgesic drugs, and the findings will provide some guidance for personalized medication. Unfortunately, the results from this study found that only 2 out of 5 patients showed some level of correlation between the results genetic analysis and clinical assessment.

Overall, this study is interesting with a lot of limitations. Major revisions need to be done before of whether or not be accepted for publishing.

Thank you for your revision of our paper. Response to your comments are provided below in red. We are sure that your suggestions can greatly improve the quality of the paper.

Major essential revisions

1)      This is human study and have to have approved protocol from Institutional Research Board for publishing the data in United States. However, it might be different in different country.

As stated in the text, the Provincial Health Ethical Review Board, consulted prior to the beginning of the study, has confirmed that it did not need authorization from the Board.

2)      What is the rational for using the saliva samples for genetic analysis?

We explained the rational for using saliva samples in the genetic analysis paragraph.

3)      Why do the samples need to stay at room temperature for 15 days? Most of the protein would degrade during the time.

We specified in the text that the sample is collected by the patient under the doctor's guidance through the kit provided and remains stable at room temperature for a maximum of 15 days, time required for sending to the laboratory and to be analysed.

4)      The authors have to provide the details about how to perform the genetic analysis.

We provided the details about how to perform the genetic analysis.

5)      The authors have to provide the results of genetic analysis of patient #1, 2, and 5 in detail.

We provided the results of genetic analyses of patients in a detailed table.

6)      Missing citations through the paper. No references from Line 49-77, Line 137-148.

We reported the missing citations in the text.

Minor revisions

1)      In Line 19, “5 tests, 2 tests” should be “5 samples, 2 samples”

We properly turned the word “5 tests” into “5 samples” (line 19) and we changed the word “2 tests… 3 tests” with “2 reports… 3 reports” in lines 19-20.  

2)      Need the product information for the Neurofarmagen Analgesia Kit

We added the product information for the Neurofarmagen Analgesia kit in Methods section.

3)      In the Table1, the “Media” should be “Mean”

We changed the word “media” with “mean”.

Reviewer 3 Report

This is an interesting study, but too premature preliminary study. The reviewer agrees with the authors that this study includes small sample size and does not include a control group. Any ideas on implication of these findings for chronic pain of multiple origins, including nerve injury, inflammation, diabetes, and chemotherapy? The introduction deserves a clear explanation on "genetic modifications implicated in the different pain's perception and response". Several data in Results are presented repeatedly/at multiple places. A detailed table can be helpful here.

Author Response

Reviewer 3

This is an interesting study, but too premature preliminary study. The reviewer agrees with the authors that this study includes small sample size and does not include a control group.

We thank you for your revision of our paper and for your interesting points regarding the potential implications of our findings. Response to your comments are provided below in red.

Any ideas on implication of these findings for chronic pain of multiple origins, including nerve injury, inflammation, diabetes, and chemotherapy?

Although the paper provides preliminary results, pharmacogenetic testing proves to be reliable when clinical conditions are not associated with comorbidities or confounding factors (e.g. polypharmacotherapy). Comorbidity was not discussed in the paper but it could be found in the extensive report of the pharmacogenetic test (Line 243).

The introduction deserves a clear explanation on "genetic modifications implicated in the different pain's perception and response".

We included an explanation of the sentences regarding the genetic modifications implicated in the different pain’s perception and response in the Introduction section.

Several data in Results are presented repeatedly/at multiple places. A detailed table can be helpful here.

We corrected the data presented repeatedly and at multiple places, as suggested by the reviewer. We added a table in which results referring to the patients are shown in details.

Round 2

Reviewer 2 Report

The authors have already addressed the major concerns.

Reviewer 3 Report

Satisfactory revisions